# Transition to the Haldane phase driven by electron-electron correlations

A. Jażdżewska[1], M. Mierzejewski [2], M. Środa [2], A. Nocera[3], G. Alvarez[4], E. Dagotto [5,6] & J. Herbrych [2]✉

One of the most famous quantum systems with topological properties, the spin $\mathcal{S}=1$ antiferromagnetic Heisenberg chain, is well-known to display exotic $\mathcal{S}=1/2$ edge states. However, this spin model has not been analyzed from the more general perspective of strongly correlated systems varying the electron-electron interaction strength. Here, we report the investigation of the emergence of the Haldane edge in a system of interacting electrons – the two-orbital Hubbard model—with increasing repulsion strength $U$ and Hund interaction $J_H$. We show that interactions not only form the magnetic moments but also form a topologically nontrivial fermionic many-body ground-state with zero-energy edge states. Specifically, upon increasing the strength of the Hubbard repulsion and Hund exchange, we identify a sharp transition point separating topologically trivial and nontrivial ground-states. Surprisingly, such a behaviour appears already at rather small values of the interaction, in a regime where the magnetic moments are barely developed.

The precise role of the electron-electron interaction in many condensed matter systems is still under much debate. From the high critical temperature superconductivity of copper- and iron-based compounds to the magnetic properties of idealized spin models, strong correlations appear crucial for our understanding of materials physics. In parallel, topology in various compounds has been typically realized and investigated at the level of non-interacting band structures in the presence of spin-orbit coupling. However, the detailed study of the Coulomb correlation effects intertwined with topological physics has barely started and represents one of the grand challenges of present-day theoretical and experimental physics.

In particular, in one of the most famous topologically nontrivial systems, i.e., the $\mathcal{S}=1$ antiferromagnetic (AFM) Heisenberg model $H_S = J\sum_\ell \mathbf{S}_\ell \cdot \mathbf{S}_{\ell+1}$ on a one-dimensional (1D) lattice geometry, the spin-spin interactions are necessary to form the zero-energy edge states, which is the hallmark of topological states. In his seminal work[1,2], Haldane showed that integer $\mathcal{S}=1,2,\ldots$ and half-integer $\mathcal{S}=1/2,3/2,\ldots$ spin systems behave fundamentally different: the former are gapped

while the latter are gapless. Affleck, Kennedy, Lieb, and Tasaki (AKLT) proved[3] that the ground-state of $\mathcal{S}=1$ chains, when generalized including biquadratic interactions, can be expressed as a valence bond state (VBS) composed of interacting $\mathcal{S}=1/2$-like singlets. In this picture, the AKLT state, when defined on an open chain, has two unpaired $\mathcal{S}=1/2$ spins at the edges of the system, forming zero-energy modes.

The existence of topologically protected edge states in $\mathcal{S}=1$ chains has been shown by extensive theoretical[4–7] and experimental[8–13] studies. Also, the road to the Haldane states from well-formed $\mathcal{S}=1/2$ spins has been studied. The AKLT VBS state initiated various investigations of extended Bose Hubbard model (containing nearest-neighbor interactions)[14,15] and ladder-like $\mathcal{S}=1/2$ systems[16–18]. In the latter, the topological $\mathcal{S}=1$ Haldane phase is a consequence of competition between various kinetic terms (i.e., ferromagnetic rung exchange or AFM frustration) or unpaired sites at the edges of overall AFM systems. Although such systems are fruitful playground for theoretical investigations and even are realized in cold atoms in optical lattice setups[11], they do not fully capture the physics of solid-state compounds. In real low-dimensional materials[19],

[1]Faculty of Physics and Astronomy, University of Wrocław, 50-383 Wrocław, Poland. [2]Institute of Theoretical Physics, Wrocław University of Science and Technology, 50-370 Wrocław, Poland. [3]Department of Physics and Astronomy and Stewart Blusson Quantum Matter Institute, University of British Columbia, Vancouver, BC V6T 1Z1, Canada. [4]Computational Sciences and Engineering Division, Oak Ridge National Laboratory, Oak Ridge, TN 37831, USA. [5]Department of Physics and Astronomy, University of Tennessee, Knoxville, TN 37996, USA. [6]Materials Science and Technology Division, Oak Ridge National Laboratory, Oak Ridge, TN 37831, USA. ✉e-mail: jacek.herbrych@pwr.edu.pl

the $\mathcal{S}=1$ moments should arise due to the electron-electron correlations in a multi-orbital Hubbard model, which is technically challenging. Because the $\mathcal{S}=1/2$ moments themselves are already an effective description of some fermionic systems, such analysis is usually unnecessary for many compounds. But in more refined descriptions, the Coulomb repulsion and Hund's coupling not only cooperate but also can compete[20,21]. Depending on their specific values, the Mott localization of electrons and the formation of well-developed spins can occur in portions of the phase diagram. As an example, in the largest family of $\mathcal{S}=1$ chains, the nickel-based compounds[19], the two $e_g$ electrons of Ni$^{+2}$ ions are necessary to form the $\mathcal{S}=1$ spins due to the Hund's rule that maximizes the on-site magnetic moment. For AgVP$_2$S$_6$ or Tl$_2$Ru$_2$O$_7$, the $\mathcal{S}=1$ spins develop, instead, on the $t_{2g}$ orbitals of V$^{+3}$ or Ru$^{+4}$, respectively. In all the previously mentioned compounds, the emergence of the topological states is unknown when described from the more fundamental perspective of quantum mechanically fluctuating individual mobile electrons, including electron-electron interaction.

To fully understand how the topological state in $\mathcal{S}=1$ chains emerges from a fermionic description, one has to focus on the effects of electron interaction within the multi-orbital systems in which Hubbard and Hund's couplings are crucial ingredients. Here, we demonstrate that these couplings are sufficient for the onset of the topologically nontrivial phase. Specifically, upon increasing the strength of the Coulomb repulsion, we identify a previously unknown transition between topologically trivial and nontrivial ground states. Our analysis unveils the threshold value of the interaction $U_c$ where the Haldane gap opens. Although at $U_c$ we also identify the emergence of zero-energy edge states and finite string order correlations (the signature properties of $\mathcal{S}=1$ Haldane phase), surprisingly, the magnetic moments are far from being fully developed, and spin excitations still resemble those in the regime of weak $U \to 0$. Consequently, we here report that the Haldane phase is not limited by having $\mathcal{S}=1$ moments. Specifically, its generalized existence can extend to unexpectedly small values of the interaction $U$-$W$, with $W$ being the kinetic energy half-bandwidth.

## Results
### From two-orbital to Heisenberg model
We employ the zero-temperature density matrix renormalization group method[4,22,23] (DMRG) to solve the 1D two-orbital Hubbard model

(2oH) at half electronic filling ($n=2$, i.e., two particles per site; one particle per orbital) and zero total magnetization $S_{tot}^z = 0$, relevant for Ni$^{+2}$-based compounds. The 2oH is given by

$$
\begin{aligned}
H_H = {} & \sum_{\gamma\gamma'\ell\sigma} t_{\gamma\gamma'}\left(c_{\gamma\ell\sigma}^\dagger c_{\gamma'\ell+1\sigma} + \text{H.c.}\right) + U\sum_{\gamma\ell} n_{\gamma\ell\uparrow} n_{\gamma\ell\downarrow} \\
& + U'\sum_\ell n_{0\ell} n_{1\ell} - 2J_H \sum_\ell \mathbf{S}_{0\ell}\cdot\mathbf{S}_{1\ell} \\
& + J_H \sum_\ell \left(P_{0\ell}^\dagger P_{1\ell} + \text{H.c.}\right).
\end{aligned}
\tag{1}
$$

Here: $\ell = 1, \ldots, L$ represents the site index, $\gamma = 0, 1$ the orbital index, and $\sigma = \uparrow, \downarrow$ the spin index. This model is generic and it can be derived from matrix elements of the fundamental $1/r$ Coulomb repulsion on the basis of atomic orbitals, following the Kanamori procedure[24]. Although challenging, the above model contains the most generic many-body interactions found in multi-orbital systems: $U$ and $U' = U - 5J_H/2$ represent the intra- and inter-orbital electron-electron Coulomb repulsion, respectively, while $J_H$ accounts for the Hund rule, i.e., ferromagnetic exchange between spins at different orbitals; finally, $P_{0\ell}^\dagger P_{1\ell}$ with $P_{\gamma\ell}^\dagger = c_{\gamma\uparrow\ell}^\dagger c_{\gamma\downarrow\ell}^\dagger$ represents the doublon-holon exchange. We will focus on degenerate bands with $t_{00} = t_{11} = t = 0.5$ [eV], $t_{01} = t_{10} = 0$, and in the following, we will use the half-bandwidth of kinetic energy as a unit, i.e., $W = 2t = 1$[eV]. While we will mostly consider the $J_H/U = 0.25$ case, other values of the Hund exchange will also be investigated (see Supplementary Note 1). Note that the $\mathbf{S}_{\gamma\ell}$ operators represent the spin-1/2 of electrons and that the above model preserves the SU(2) symmetry provided that $U' = U - 5J_H/2$ and the doublon-holon exchange term is included[25].

The standard probe of spin excitations is the momentum $q$ and energy $\omega$ resolved dynamical spin structure factor $S(q, \omega)$, which is the Fourier transform of the non-local Green's functions $\langle\langle \mathbf{T}_\ell \mathbf{T}_{\ell'}\rangle\rangle_\omega$ (see Methods), with $\mathbf{T}_\ell$ as the total on-site spin $\mathbf{T}_\ell = \sum_\gamma \mathbf{S}_{\gamma\ell}$. The calculated $S(q, \omega)$ is routinely compared to inelastic neutron scattering (INS) or resonant inelastic X-ray scattering data, also in the case of $\mathcal{S}=1$ compounds. With increasing strength of interaction $U$, the 2oH spectrum (Fig. 1A) develops from a continuum of $\mathcal{S}=1/2$-like excitations at $U=0$[26,27] to the well-established magnon-like excitations[28,29] of the $\mathcal{S}=1$

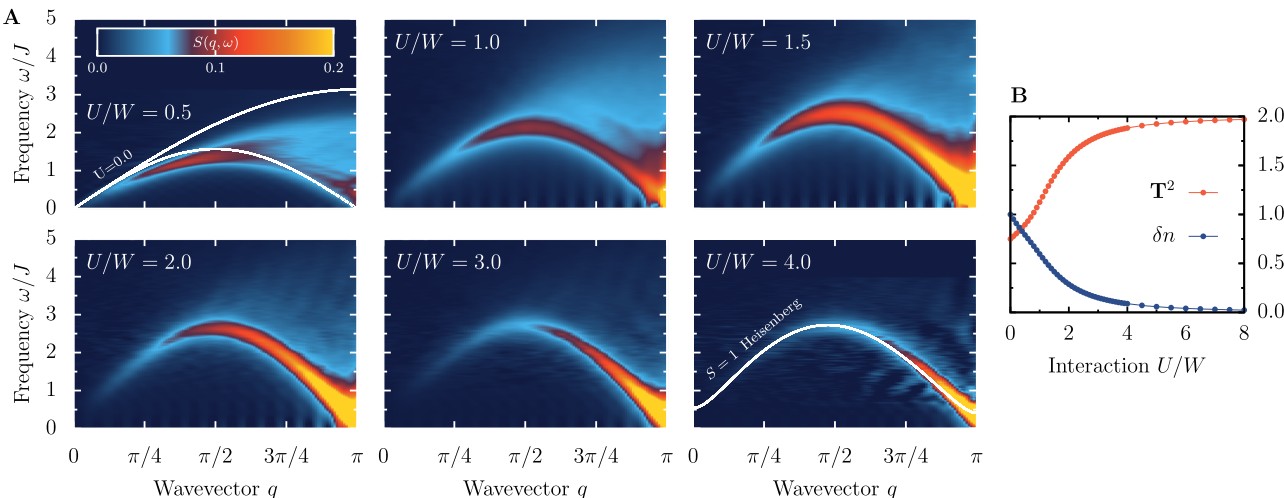

**Fig. 1 | Spin excitations. A** Evolution of the spin excitations, as measured by the dynamical spin structure factor $S(q, \omega)$, with increasing strength of electron-electron interaction $U$ for a system of $L=80$ sites and $J_H/U = 0.25$. The frequency scale was renormalized by the effective spin exchange $J = 2t^2/(U + J_H)$. White lines in the left top panel represent the two-spinon continuum of the $U=0$ Hubbard model, while the line in the bottom right panel depicts the magnon dispersion of the $\mathcal{S}=1$ Heisenberg model. In the open boundary systems considered here, the zero-energy Haldane edge states are expected at $\omega=0$. However, the large intensity of this modes can blur the spectra's details. To avoid this issue, we have evaluated the spin excitations only in the bulk of the system (see Methods). **B** Total magnetic moment per site $\mathbf{T}^2 = \mathcal{S}(\mathcal{S}+1)$ and charge fluctuations $\delta n$ vs. interaction strength $U$. Note $\mathbf{T}^2$ starts at 0.75 for non-interacting $U=0$ electrons.

**A**

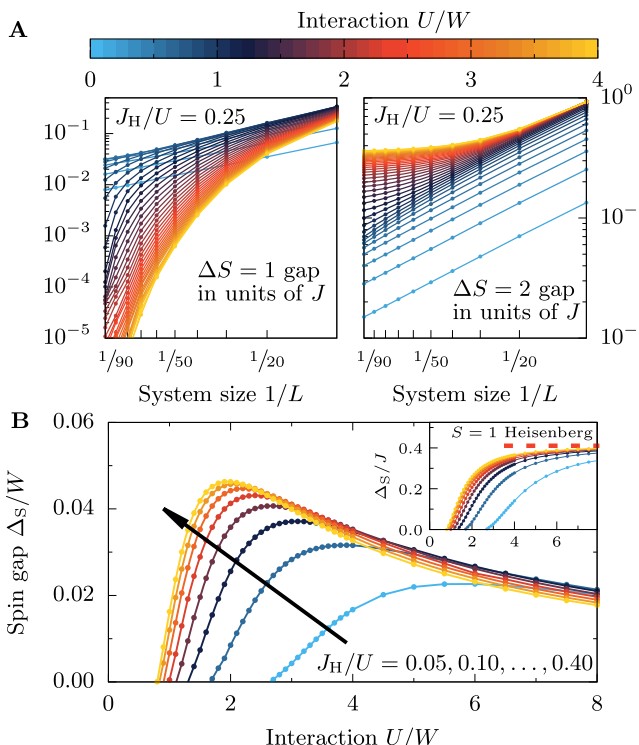

**B**

**Fig. 2 | Spin gaps. A** Finite-size scaling of $\Delta S = 1$ (left panel) and $\Delta S = 2$ (right panel) spin excitations for $J_H/U = 0.25$ and $L \in \{10, 20, ..., 100\}$. Line color code represents the value of the interaction $U$. **B** $U$ dependence of the extrapolated magnon gaps in units of $W$. Top to bottom: $J_H/U = 0.05, 0.10, ..., 0.40$. Inset depicts the same data but renormalized by the effective spin exchange $J$. The saturation to the Haldane gap $\Delta_S/J \simeq 0.41$ is clearly visible (red dashed line).

Heisenberg model at large $U \gg W$. Renormalizing the frequency by the effective spin exchange, $J = 2t^2/(U + J_H)$[20], yields qualitative agreement between the models at $U/W \simeq 4$. As expected, for such value of interaction $U$, the average total magnetic moment is almost maximized $\mathbf{T}^2 = \mathcal{S}(\mathcal{S}+1) \simeq 2$ and the charge fluctuations $\delta n = \langle n^2 \rangle - \langle n \rangle^2$ are vanishing (Fig. 1B).

The artificial broadening needed in the dynamical-DMRG method[30,31], which physically mimics the influence of disorder, finite temperature, and measurements-device resolution, prevents us from extracting accurate values of the magnon gap directly from the spectrum of $S(q, \omega)$. Instead, the gaps can be obtained from the difference in ground-state energies of two magnetization sectors with different $S_{tot}^z$ (with $\Delta S$ being the magnetization difference) at fixed electron density $n$. It is important to note that when working on a finite-size lattice, the $\Delta S = 1$ excitations of 2oH are always gapless when extrapolated to the thermodynamic limit $L \to \infty$ (Fig. 2A). For $U \to 0$, the gapless spin excitations manifest the physics of non-interacting fermions, with an inverse-linear dependence on the system size $\mathcal{O}(1/L)$ of the gap according to Lieb-Schultz-Mattis theorem[32]. In the opposite limit of the $\mathcal{S} = 1$ Heisenberg model at $U \gg W$, the gapless $\Delta S = 1$ excitation originates in a four-fold degenerate ground-state (two-fold in the $S_{tot}^z = 0$ sector) with two $\mathcal{S} = 1/2$ edge states[29,33]. For a finite $L$, these edge states are split due to their overlap[34], which decays exponentially with increasing system size. See large $U$ data in Fig. 2A. Thus, within the open boundary condition system with edge states, the true magnon gap $\Delta_S$ can be extracted from $\Delta S = 2$ excitations[4,35,36]. Still, for $U \to 0$, the magnons are gapless with $\mathcal{O}(1/L)$ size dependence of the gap.

On the other hand, increasing the strength of $U$ changes the nature of the scaling. At large $U$, we observe a saturation to a finite value in the $L \to \infty$ limit. This saturation is to the well-known Haldane gap $\Delta_S/J \simeq 0.41$ for $U \gtrsim 4$, confirming the accuracy of our procedure. Crucially, the finite-size scaling varying $U$ reveals a novel critical (Hund $J_H$ dependent, see Supplementary Note 1) value of the interaction

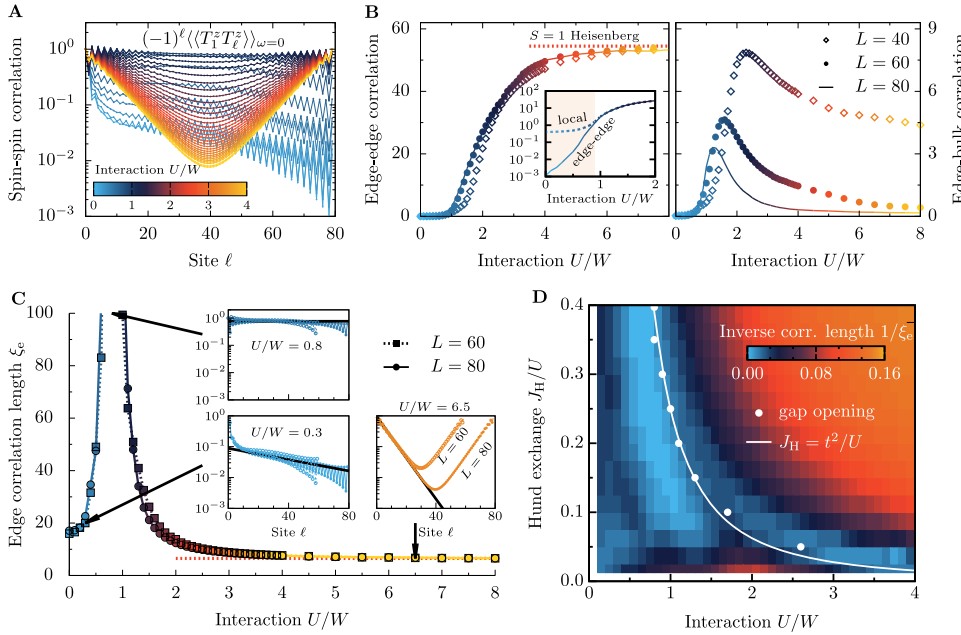

**Fig. 3 | Edge spin correlations. A** Distance $\ell$ dependence of the zero-frequency $\omega = 0$ dynamical spin-spin correlations $(-1)^\ell \langle\langle T_1^z T_\ell^z \rangle\rangle_{\omega=0}$ for various values of interaction $U$ (denoted by color code). The results are normalized by the $\ell = 1$ value of the correlation function. **B** Edge-edge $|\langle\langle T_1^z T_L^z \rangle\rangle_{\omega=0}|$ (left panel) and edge-bulk $|\langle\langle T_1^z T_{L/2}^z \rangle\rangle_{\omega=0}|$ (right panel) dynamical spin correlations vs. interaction strength $U$. At $U_c$, we observe the appearance of finite edge-edge correlations, saturating at $U \gg W$ to the value given by the $\mathcal{S} = 1$ Heisenberg model (red dashed line).

**C** Extracted, Eq. (2), edge correlation length $\xi_e$ vs. interaction strength $U$. Insets depict examples of spin-spin correlations for two system sizes ($L = 60$ and $L = 80$, together with fitted exponentials $\propto \exp(-\ell/\xi_e)$. All data are calculated at $J_H/U = 0.25$. **D** Interaction $U/W$–Hund exchange $J_H/U$ phase diagram on the basis of inverse edge correlation length $1/\xi_e$ for $L = 60$. White points depict $U_c$ obtained from the spin gap $\Delta_S$ opening, while the white line represents $J_H = t^2/U$.

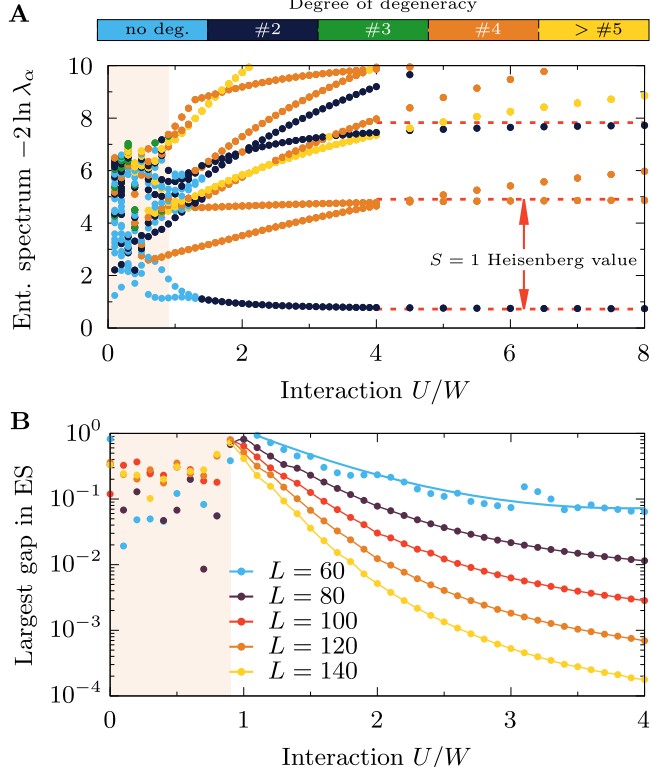

**Fig. 4 | Topological phase transition. A** Interaction $U$ dependence of the entanglement spectrum $-2 \ln \lambda_\alpha$, obtained at $J_H/U = 0.25$ using a $L = 140$ site system partitioned in half. Color code depicts the number of occurrences of a given eigenvalue (number of degeneracies). The values for the $\mathcal{S} = 1$ Heisenberg model are also displayed (red dashed lines). **B** Analysis of the largest gap in the entanglement spectrum for various system sizes $L = 60, 80, 100, 120, 140$ (see also Supplementary Note 3).

$U_c = U_c(J_H)$ where the gap opens (Fig. 2B). For example, for $J_H/U = 0.25$, the magnons become gapped at $U_c/W \simeq 0.9$.

A remarkable result of our computational investigations is that the magnon gap $\Delta_S$ opens at a value of the interaction $U = U_c$ for which the overall spin excitations are *far* from the $\mathcal{S} = 1$ Heisenberg model magnon-like spectrum. In fact, for $U/W \sim 1$, the spin excitations still visually resemble the non-interacting continuum of $\mathcal{S} = 1/2$-like moments, though with redistributed spectral weights (Fig. 1A).

### Zero-energy edge modes

As mentioned, the exponential in the system size dependence of the $\Delta S = 1$ gaps (Fig. 2A) indicates the presence of edge states. To quantify them, we analyze (Fig. 3) the zero-frequency $\omega = 0$ dynamical spin-spin correlation functions between the edge and the bulk of the system, i.e., the non-local Green's functions $(-1)^\ell \langle\langle T_1^z T_\ell^z \rangle\rangle_{\omega=0}$, capable of capturing zero-energy modes. Here, the $(-1)^\ell$ prefactor removes the AFM staggered pattern. At small $U$, the spin correlations decay exponentially with distance $\ell$ (Fig. 3A), as expected for a paramagnetic region. Increasing $U$ leads to a slower decay, although still exponential. At $U \simeq U_c$, the $\omega = 0$ correlations are approximately site-independent. Note that this does not originate in any long-range order because the value of spin correlations decays with the system size (see Fig. 3B and the discussion below).

Interestingly, a characteristic V-shape of correlations develops above $U_c$. This is the manifestation of the edge states present at the (open) boundaries of the system[5]. In the $\mathcal{S} = 1$ Heisenberg model, the zero-energy modes are not localized at a single edge site but decay exponentially with the correlation length $\xi_S \simeq 6.1$. This leads to finite (exponentially suppressed) AFM spin correlations up to half $\ell \sim L/2$ of

the system. The increase of $\langle\langle T_1^z T_\ell^z \rangle\rangle_{\omega=0}$ for $\ell > L/2$ is exactly a consequence of correlated edge states: the edge-edge correlations are finite, while the edge-bulk correlations are vanishing.

To assess the development of spin-spin correlations in the 2oH system, especially the correlated edge states, we monitor the behavior of the edge-edge and edge-bulk (Fig. 3B) values vs. the interaction $U$. The edge-edge correlations acquires a non-zero value at $U_c$ (see Supplementary Note 2) and displays small finite-size effects. On the other hand, the finite value of the edge-bulk correlations decreases with system size $L$ and vanishes in the $L \to \infty$ limit.

Furthermore, we can extract the interaction dependence of the edge correlation length (Fig. 3C) by fitting $\ell < L/2$ data of the 2oH to

$$(-1)^\ell \langle\langle T_1^z T_\ell^z \rangle\rangle_{\omega=0} \propto \exp(-\ell/\xi_e). \tag{2}$$

For $U/W > 4$ we reproduce $\xi_e \simeq \xi_S \simeq 6.1$, consistent with dynamical spin structure factor $S(q, \omega)$ investigations of the $\mathcal{S} = 1$ Heisenberg model physics. Interestingly, the extracted $\xi$ diverges at $U_c$. This divergence reflects the site-independent correlations in this region (see Supplementary Note 2).

### Topological phase transition

The opening at $U_c$ of a spin gap $\Delta_S$, the emergence of edge-edge correlations $\langle\langle T_1^z T_\ell^z \rangle\rangle_{\omega=0}$, and the diverging edge correlation length $\xi_e$ all consistently indicate the existence of an interaction-induced topological phase transition between topologically trivial and nontrivial regions, with the emergence of the Haldane edge states at $U_c$. The topological phases can be identified by investigating the entanglement spectrum of the system[37,38], i.e., the Schmidt coefficients $\lambda_\alpha$ of left/right ($|L\rangle/|R\rangle$) decomposed ground-state $|gs\rangle = \sum_\alpha \lambda_\alpha |L\rangle_\alpha |R\rangle_\alpha$, with $\lambda_\alpha^2$ being the eigenvalues of the reduced density matrix of the partition. In the topologically nontrivial region, all $\lambda_\alpha$'s are evenly degenerate. Consequently, the entanglement entropy $S_{vN} = -\sum_\alpha \lambda_\alpha^2 \ln \lambda_\alpha^2$ cannot drop below the $\ln 2$ value for any cut of the system, consistent with the presence of entangled $\mathcal{S} = 1/2$ edge states. The analysis of the 2oH model indicates that this condition is fulfilled for $U \gtrsim U_c$ (Fig. 4A). Detailed investigation of the largest gap (see Supplementary Note 3) in the entanglement spectrum (Fig. 4B) shows that the trivial region $U < U_c$ does not have any apparent structure in the $\lambda_\alpha$ eigenvalues. On the other hand, the largest gap decays exponentially with system size for any $U > U_c$ (though, with slower decay in the proximity of $U_c$) and vanishes in the thermodynamic limit $L \to \infty$.

In the context of the $\mathcal{S} = 1$ Heisenberg model, the topological Haldane phase can also be detected by studying the non-local string order parameter[33,39,40]

$$\mathcal{O}_s(\ell) = -\left\langle A_m \exp\left(i\theta \sum_{n=m+1}^{m+\ell-1} A_n\right) A_{m+\ell} \right\rangle, \tag{3}$$

which for $\theta = \pi$ and $A_\ell = S_\ell^z$ measures the breaking of the discrete $Z_2 \times Z_2$ hidden symmetry (i.e., the dihedral group of $\pi$ rotations). It is important to note that the phase $\theta = \pi$ was obtained via the valence bond state structure of the AKLT state. For a generic spin-$\mathcal{S}$ Heisenberg model, the string order phase becomes spin-dependent $\theta = \theta(\mathcal{S})$, i.e., it has to reflect the properties of a given VBS ground-state[41-44].

In the case of the 2oH model, for $U > U_c$, the $\pi$-string order $\mathcal{O}_s$ does not decay (Fig. 5), as expected in the $\mathcal{S} = 1$ Haldane phase. However, it is important to note that the total spin operator of 2oH, $A_\ell = T_\ell^z$, involves not only $\mathcal{S} = 1$ but also $\mathcal{S} = 1/2$ degrees of freedom and that for $U \simeq U_c$ the magnetic moment deviates strongly from $\mathcal{S} = 1$ (Fig. 1B). Nevertheless, we observe a finite string order all the way down to $U = U_c \sim W$, showing that this type of order can exist in a fermionic system as well, even without well-defined moments. Interestingly, consistent with the topological phase transition at $U_c$, for $U < U_c$ the string order vanishes, and the system size dependence changes from weakly increasing with

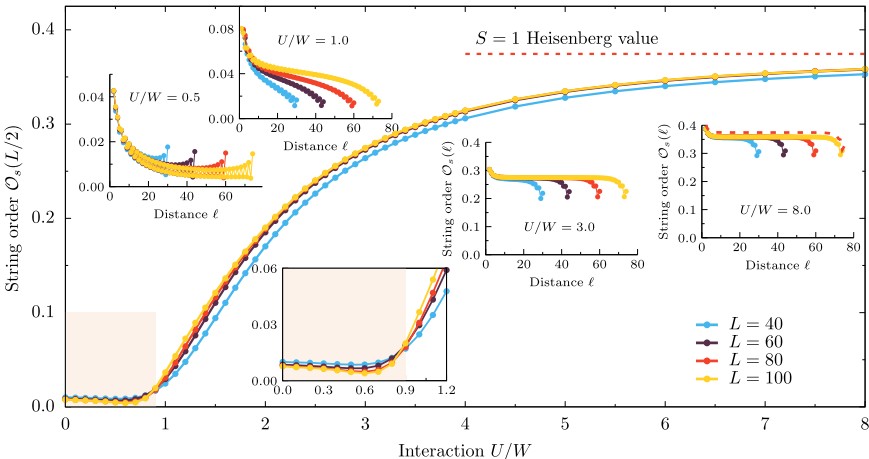

**Fig. 5 | String order.** Interaction $U$ dependence of the string order parameter $\mathcal{O}_c(\ell)$ with $\theta = \pi$ phase at $\ell = L/2$ distance in bulk ($m = L/4$). Upper insets depicts $\mathcal{O}_c(\ell)$ vs. distance $\ell$ for $U = 0.5, 1.0, 3.0, 8.0$ (left to right). The lower inset depicts a zoom to the proximity of the phase transition $U_c$, with the shaded region depicting the trivial phase. All data are evaluated at $J_H/U = 0.25$ using $L = 40, 60, 80, L = 100$ site systems.

$L$ (for $U > U_c$) to weakly decreasing with $L$ (for $U < U_c$). The latter is consistent with the slow scaling of $\mathcal{O}_s$ for $S = 1/2$ moments[45].

## Discussion

The non-local character of the topological states allows for phase transitions even in 1D (rare phenomena due to the Mermin-Wagner theorem). Our numerical results indicate that the correlated one-dimensional two-orbital Hubbard model has a sharp transition at $U_c \sim W$ between a topologically trivial region and a generalized fermionic Haldane phase with edge states. Surprisingly, the magnetic moments are not yet fully developed in a vast region of the topological phase (Fig. 1B), and thus the $S = 1$ Heisenberg model-like description cannot be applied directly, and it is not necessary to describe the physics of the fermionic generalized Haldane phase presented here. Actually, our analysis shows that the gapped ground-state with finite string order survives down to $U \sim W \sim \mathcal{O}(t)$. Consequently, this result indicates that a VBS-like state, similar to the AKLT state, could be formulated[46] even with mobile fermions. It seems true despite the fact that the length scale of spin-spin correlations indicates the spatially extended character of the ground-state, although with moments small in value. Our detailed interaction $U$ and Hund exchange $J_H$ investigation (Fig. 3D) indicates that the SU(2) symmetric system undergoes the transition at $J_H \simeq t^2/U$, and consequently a finite $U \sim W$ is necessary for the onset of the non-topological–topological phase transition in real materials.

Furthermore, our results indicate that the details of the band structure, i.e., of the hopping matrix $t_{\gamma\gamma'}$, are not crucial for our findings. Up to now, we have considered degenerate bands, i.e., $t_{00} = t_{11} = 0.5$ [eV] and $t_{01} = t_{10} = 0$. In Fig. 6 we present additional results of edge correlation length $\xi_e$ (discussed in Fig. 3) and string order parameter $\mathcal{O}_s(L/2)$ (discussed in Fig. 5) for non-degenerate bands ($t_{00} = 0.5$ [eV], $t_{11} = 0.3$ [eV], $t_{01} = t_{10} = 0$, with $W = 1$ [eV]) and strongly hybridized orbitals ($t_{00} = 0.5$ [eV], $t_{11} = 0.3$ [eV], $t_{01} = t_{10} = 0.5$ [eV], with $W = 1.8$ [eV]). For all considered cases, we find the transition (identified by diverging $\xi_e$ and the onset of non-zero $\mathcal{O}_s$) to the Haldane phase at a finite value of interaction $U$. Consequently, our results are relevant for various low-dimensional $S = 1$ compounds, irrespective of kinetic energy details, i.e., for recently investigated platforms such as the van der Waals oxide dichlorides MOX$_2$ (M=V, Ta, Nb, Ru, Os, and X = halogen element)[47] or metal-organic structures[48]. Another promising candidate to test the prediction of our work is the van der Waals quasi-1D material OsCl$_4$[49].

Also, one could expect that for $J_H \gg U$ (i.e., when the system always has well-developed on-site triplets formed by electrons), even

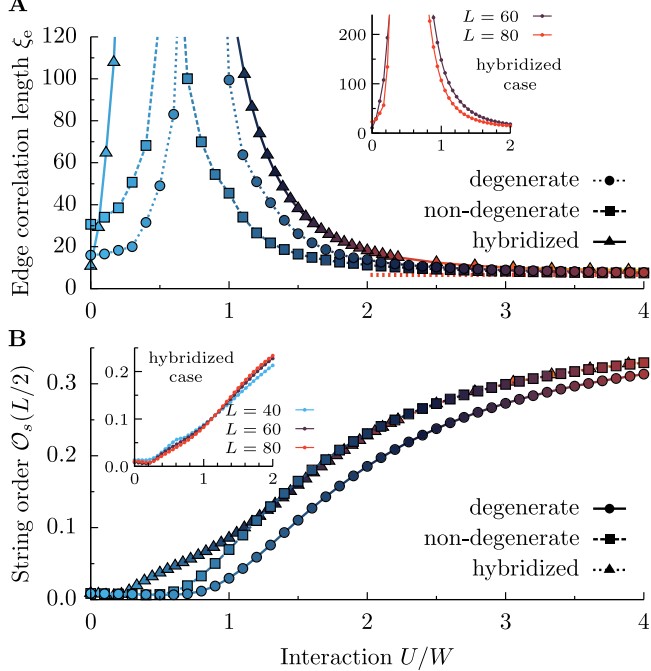

**Fig. 6 | Band structure dependence.** Interaction $U/W$ dependence of **A** edge correlation length $\xi_e$ and **B** string order parameter $\mathcal{O}_s(L/2)$ for degenerate, non-degenerate bands, and also for the strongly hybridized case (see text for details). Main panels depict $L = 60$ and $J_H/U = 0.25$ data, while insets depict finite-size scaling of strongly hybridized cases.

small interaction will induce the Haldane phase. However, such region of parameter space is unrealistic because for $J_H/U > 0.4$ the inter-orbital interaction $U' = U - 5J_H/2$ becomes attractive $U' < 0$. It is therefore evident that setups with coupled $S = 1/2$ triplets represent, from the electron system perspective, broken spin rotation with $U' \neq U - 5J_H/2$. Previous analysis of the Haldane phase in such setups indicates its fragility with respect to charge fluctuations[16–18]. Our results indicate that within a two-orbital setup, the Haldane phase is robust down to rather small values of the interaction $U$, in a regime where the magnetic moments are barely developed. Thus, our results are generalizing the ideas of Haldane for $S = 1$ spin Heisenberg models into unexplored territory involving delocalized electrons. The structure of Haldane edge states was previously investigated, e.g., via INS experiments[10].

However, our results indicate that the energy gap separating the edge modes from the magnon-like excitations can be small (even exponentially small at the transtion $U \sim U_c$, see Fig. 2B). Similarly, the intensity of such modes is diminishing close to the transition (see Fig. 3B). As a consequence, neutron scattering (as a global probe of the sample) would not necessarily be the best tool. An alternative would be local probes, e.g., nuclear magnetic resonance experiments[50], exploiting the large edge correlation length $\xi_e \gg \xi_S$ (quantified by the decay of staggered magnetization at the edges of the system).

## Methods

### DMRG method

The Hamiltonians and observables discussed here were studied using the zero-temperature DMRG method[4,23] within the single center site approach[22], where the dynamical correlation functions are evaluated via the dynamical-DMRG[30,31], i.e., calculating spectral functions directly in frequency space with the correction-vector method using the Krylov decomposition[31]. We have kept up to $M = 3072$ states, performed at least 15 sweeps, and used $A = 0.001$ vector-offset in the single-site DMRG approach, allowing to accurately simulate system sizes up to $L \lesssim 140$ sites of the two-orbital Hubbard model. Consequently, the error bars on the numerical results are smaller than the data points.

### Dynamical spin structure factor

The dynamical spin structure factors are evaluated as

$$S(q,\omega) = \frac{2}{L+1} \sum_{\ell=1}^{L} \cos\left[(\ell - L/2)q\right] \langle\langle \mathbf{T}_\ell \mathbf{T}_{L/2} \rangle\rangle_\omega, \qquad (4)$$

where $q = n\pi/(L+1)$, $n = 0, \ldots, L$, and non-local Green's function is given by

$$\langle\langle \mathbf{T}_m \mathbf{T}_n \rangle\rangle_\omega = -\frac{1}{\pi} \operatorname{Im} \left\langle \mathrm{gs} \middle| \mathbf{T}_m \frac{1}{\omega + i\eta - H + \epsilon_0} \mathbf{T}_n \middle| \mathrm{gs} \right\rangle. \qquad (5)$$

Here $|\mathrm{gs}\rangle$ represents the ground-state with energy $\epsilon_0$. The $S(q, \omega)$ spectra presented in Fig. 1A of the main text were calculated with the frequency resolution $\delta\omega/J \simeq 0.03$ and broadening $\eta = 2\delta\omega$ [note the $U$ dependence of the spin exchange $J = 2t^2/(U + J_H)$].

### Largest gap in the entanglement spectrum

In order to find the largest gap in the entanglement spectrum, first we have calculated consecutive gaps $\delta_n = \min(\ln \lambda_n - \ln \lambda_{n-1}; \ln \lambda_{n+1} - \ln \lambda_n)$. The largest gap is then obtained from $\max(\delta_1, \delta_2, \ldots)$.

## Data availability

The data generated in this study have been deposited in the Zenodo database under the accession code https://doi.org/10.5281/zenodo.7854617.

## Code availability

We have used the DMRG++ computer program developed at Oak Ridge National Laboratory. The code that supports this study is available at the Oak Ridge National Laboratory repository https://code.ornl.gov/gonzalo_3/dmrgpp. The input scripts for the DMRG++ package to reproduce our results can be found at https://bitbucket.org/herbrychjacek/corrwro/ and on the DMRG++ webpage.

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

## Acknowledgements

M.M. acknowledges support from the National Science Centre (NCN), Poland, via project 2020/37/B/ST3/00020. M.Ś. and J.H. acknowledge grant support from the National Science Centre (NCN), Poland, via project 2019/35/B/ST3/01207. A.N. acknowledges support from the Max Planck-UBC-UTokyo Center for Quantum Materials and Canada First Research Excellence Fund (CFREF) Quantum Materials and Future Technologies Program of the Stewart Blusson Quantum Matter Institute (SBQMI), and the Natural Sciences and Engineering Research Council of Canada (NSERC). G.A. was partly supported by the Scientific Discovery through Advanced Computing (SciDAC) program funded by the U.S. DOE, Office of Science, Advanced Scientific Computing Research and BES, Division of Materials Sciences and Engineering. E.D. was supported by the U.S. Department of Energy, Office of Science, Basic Energy Sciences, Materials Sciences and Engineering Division. Part of the calculations have been carried out using resources provided by Wroclaw Centre for Networking and Supercomputing.

## Author contributions

J.H. conceived the study. A.J., M.M., E.D., and J.H. planned the project. A.J., M.Ś., and J.H. performed the numerical experiments and analyzed the data. A.N. and G.A. developed and tested the simulation codes. M.M., E.D., and J.H. wrote the manuscript. All authors provided comments on the publication.

## Competing interests

The authors declare no competing interests.
