## [Peer Review File · Nature Communications]

REVIEWER COMMENTS

Reviewer #1 (Remarks to the Author):

This paper explores the topological phase transitions of the two-orbital Hubbard chain, parameterized by two parameters the Hubbard interaction U and a ferromagnetic Hund exchange interaction J_H . When U and J_H are large, the model reduces to the spin-1 Heisenberg model which realizes a gapped SPT phase with gapless edge modes. Their key finding is an interaction-induced continuous phase transition as a function of U and J_H , which occurs in a regime where charge fluctuations are strong and where local $S=1$ moments are not well formed. The paper is interesting and solid, and the results are convincing. I just have a few minor comments that may help to improve the paper.

1. In the intro, the sentence:

“In his semi-

nal work [1, 2], Haldane showed that integer $S = 1, 2, \dots$

and half-integer $S = 1/2, 3/2, \dots$ spin systems behave

fundamentally different: the latter are gapped while the

former are gapless.”

seems like a typo, since it is the integer S which are gapped and half-integer which are gapless (the sentence says the opposite).

I also noticed multiple weird uses of the word “latter” throughout the text, which I found confusing.

2. In Eq 1, the indices γ , ℓ , σ should be defined.

3. Can the authors provide citations for the Hamiltonian Eq 1? It would also benefit to explain somewhere why $U' = U - 5J_H/2$ is the appropriate value to preserve $SU(2)$ symmetry, and also why the hopping term the same amplitude between same and different orbitals.

4. In Fig2B, J_H/U takes on values 0.05 ... 0.4, which should be a total of 8 values. However, I only count seven datasets shown on the plot, so it is not clear which color corresponds to which value of J_H/U .

5. Fig3, is it clear that the edge-bulk correlations should decay exponentially in the small U phase? Could it instead be decaying algebraically? Relatedly, at the critical point, how do the authors know if the correlations are truly "site independent", or if they are just decaying very slowly?

Reviewer #2 (Remarks to the Author):

This paper studies the emergence of a Haldane phase in a two-orbital Fermi-Hubbard model. Through tuning the strength of Hubbard interaction, the authors observe a transition between topologically trivial and non-trivial phases, witnessed by several typical features including the presence of gapped excitation and edge states, a change on the string order parameter, and a gap closing in the entanglement spectrum. The results are clear, however, in my opinion, not significant enough for publishing on Nature Communications.

The main finding of this paper is the emergence of Haldane phase from an interacting electronic system instead of spin chain, however, this is not a new result. For example, Ref.6 and Ref.7 reported such emergence in a very similar two-orbital Fermi-Hubbard model (only different at a Hund term, which should not affect on the presence of a topological transition), and presented detailed analysis on how the trivial phase is adiabatically connected to the Haldane phase. The authors also highlight the non $S=1$ total spin in the emergent Haldane phase with finite U, but this can be adiabatically connected to the large U case with $S=1$ total spin and the standard non-order measures do capture the topological nature. In short, the current numerical observation and underlying physics have no significant difference with existing literatures.

Besides, although the authors claim their calculation is "technically challenging", solving such a two-orbital model (two-leg ladder) is not hard for DMRG due to its quasi-1D nature. The calculation of the dynamical structure factor based on the dynamical DMRG is also standard. In the discussion part, the authors state again "investigating systems on finite lattices, especially with many-body interactions incorporated, is always a challenge...". However, as the DMRG allows accessing very large system size (say up to thousands of sites) for (quasi-)1D systems, this is not an issue for the current calculation.

Overall speaking, the authors indeed provide faithful results, and this paper should be published somewhere. However, it is not of significant importance and novelty for publication on Nature Communications.

Reviewer #3 (Remarks to the Author):

this manuscript is devoted to a study of a one-dimensional model of interacting electrons that generalizes the Hubbard model to a two-orbital case.

This is the simplest possible model with itinerant electrons that admits as a simple limit the spin-1 Heisenberg model. It thus allows to study the fate of the Haldane phase when electrons are itinerant. This is motivated by real magnets including Nickel ions for example as spin-1 localized moments.

The authors use the DMRG algorithm to give an in-depth study of this model and they are able to characterize the extent and robustness of the Haldane phase. The results are interesting and obtained by state of the art theoretical analysis.

I am inclined to recommend for publication in Nature Communications.

However I am a bit disappointed by the following point: the authors claim to study in essence a more realistic depiction of real-world Haldane magnets - they correctly mention Nickel, Vanadium and Ruthenium ions - but their conclusions are seemingly only for hard-core theorists.

No feedback is given concerning the impact of their results in present or future compound synthesis or experiments. I think that adding such arguments would strengthen the manuscript.

I appreciate the fact that the manuscript is beautifully prepared, very well written and easy to read.

Reviewer #4 (Remarks to the Author):

Reply to Referee #1:

This paper explores the topological phase transitions of the two-orbital Hubbard chain, parameterized by two parameters the Hubbard interaction U and a ferromagnetic Hund exchange interaction J_H . When U and J_H are large, the model reduces to the spin-1 Heisenberg model which realizes a gapped SPT phase with gapless edge modes. Their key finding is an interaction-induced continuous phase transition as a function of U and J_H , which occurs in a regime where charge fluctuations are strong and where local $S=1$ moments are not well formed. The paper is interesting and solid, and the results are convincing. I just have a few minor comments that may help to improve the paper.

We thank the Referee for carefully evaluating our work and for a very positive assessment. Below, we address the specific issues raised by the Referee.

1. In the intro, the sentence: “In his seminal work [1, 2], Haldane showed that integer $S = 1, 2, \dots$ and half-integer $S = 1/2, 3/2, \dots$ spin systems behave fundamentally different: the latter are gapped while the former are gapless.” seems like a typo, since it is the integer S which are gapped and half-integer which are gapless (the sentence says the opposite).

We thank the Referee for pointing out this obvious typo. Our apologies. We have corrected it in the revised version of the manuscript.

I also noticed multiple weird uses of the word “latter” throughout the text, which I found confusing.

In the revised version of the manuscript, we cleaned up several sentences to make the text more readable. This certainly has improved the quality of the paper.

2. In Eq 1, the indices γ , ℓ , σ should be defined.

In the revised version of the manuscript, we have defined all indices of Eq. (1).

3. Can the authors provide citations for the Hamiltonian Eq 1? It would also benefit to explain somewhere why $U'=U-5J_H/2$ is the appropriate value to preserve $SU(2)$ symmetry, and also why the hopping term the same amplitude between same and different orbitals.

Following the Referee's advice, we have included the new Reference 26 [A. Georges *et al.*, Annu. Rev. Condens. Matter Phys. **4**, 137 (2013)], which contains a detailed discussion on symmetries and the role of Hund coupling in the multi-orbital Hubbard model (see Section 2 in the publication above).

Concerning the details of the kinetic energy term, i.e., the amplitudes of the hopping term: in the revised version of the manuscript, we have added a new paragraph and figure discussing this issue (this issue was raised also by Referee #3). In short, our results clearly show that the

details of the hopping matrix do not play a significant role. We have investigated the degenerate bands (discussed in the previous version of the manuscript)

$$t = t_{\gamma\gamma'} = \begin{pmatrix} -0.5 & 0 \\ 0 & -0.5 \end{pmatrix},$$

and now also we discuss the non-degenerate case

$$t = t_{\gamma\gamma'} = \begin{pmatrix} -0.5 & 0 \\ 0 & -0.3 \end{pmatrix},$$

as well as the strongly hybridized case

$$t = t_{\gamma\gamma'} = \begin{pmatrix} -0.5 & -0.5 \\ -0.5 & -0.3 \end{pmatrix}.$$

Here γ represents an orbital index. In all these examples, we found that the transition from the non-topological to the Haldane phase occurs at $U \sim W$ (where W is a half-bandwidth of the kinetic energy for each given case, namely $W = 1, 1, 1.8$, respectively). Please see the new Fig. 6 and the new associated paragraph in the *Discussion and conclusion* section.

4. In Fig2B, J_H/U takes on values 0.05 ... 0.4, which should be a total of 8 values. However, I only count seven datasets shown on the plot, so it is not clear which color corresponds to which value of J_H/U .

We thank the Referee for pointing this out. In the revised version of the manuscript, we have corrected this figure, i.e., we present 8 curves for $J_H/U=0.05,0.10,0.15,0.20,0.25,0.30,0.35,0.40$.

5. Fig3, is it clear that the edge-bulk correlations should decay exponentially in the small U phase? Could it instead be decaying algebraically? Relatedly, at the critical point, how do the authors know if the correlations are truly “site independent”, or if they are just decaying very slowly?

Our numerical data indicate that the decay of edge correlations, even below the transition, can be better described by an exponential function than by a power law. In the figures below, we present examples of results on the left-hand side of the transition ($U/W = 0.6$), i.e., static and dynamic spin-spin correlations between the edge and the rest of the system, $(-1)^\ell \langle T_1^z T_\ell^z \rangle$ and $(-1)^\ell \langle \langle T_1^z T_\ell^z \rangle \rangle_{\omega=0}$, respectively. Note that these are the same data as presented in Fig.3(A) and Fig S2(A).

As it is evident from the results presented in the log-log scale (bottom row), the data are not a straight line (as expected for a power-law dependence). Our data are best described by exponential functions (see the top row of the figures below). This is especially true for the static correlations, while the dynamical ones exhibit exponential behavior at large distances $\ell > L/2$.

Concerning the behavior in the proximity of the transition ($U \sim W$), we cannot exclude an extremely weak dependence on distance (e.g., due to AFM-like oscillations visible in all our data; see Fig.3(A) and Fig S2(A)). In fact, from finite-size chain numerical results, one cannot expect *perfect site-independent* behavior. Nevertheless, our results indicate that the distance dependence of the edge spin correlations is extremely weak at the transition (as exemplified by the diverging edge correlation length $\xi_e \gg L$).

Distance dependence of spin-spin correlations between edge and the rest of the system.

Left column: static correlations $(-1)^{\ell} \langle T_1^z T_{\ell}^z \rangle$. Right column: dynamic spin correlations $(-1)^{\ell} \langle \langle T_1^z T_{\ell}^z \rangle \rangle_{\omega=0}$.

Top row: log-linear plot. Bottom row: log-log plot. Parameters: $L = 80, U/W = 0.6, J_H/U = 0.25$.

Reply to Referee #2:

This paper studies the emergence of a Haldane phase in a two-orbital Fermi-Hubbard model. Through tuning the strength of Hubbard interaction, the authors observe a transition between topologically trivial and non-trivial phases, witnessed by several typical features including the presence of gapped excitation and edge states, a change on the string order parameter, and a gap closing in the entanglement spectrum. The results are clear, however, in my opinion, not significant enough for publishing on Nature Communications.

The main finding of this paper is the emergence of Haldane phase from an interacting electronic system instead of spin chain, however, this is not a new result. For example, Ref.6 and Ref.7 reported such emergence in a very similar two-orbital Fermi-Hubbard model (only different at a Hund term, which should not affect on the presence of a topological transition), and presented detailed analysis on how the trivial phase is adiabatically connected to the Haldane phase. The authors also highlight the non $S=1$ total spin in the emergent Haldane phase with finite U , but this can be adiabatically connected to the large U case with $S=1$ total spin and the standard non-order measures do capture the topological nature. In short, the current numerical observation and underlying physics have no significant difference with existing literatures.

Besides, although the authors claim their calculation is “technically challenging”, solving such a two-orbital model (two-leg ladder) is not hard for DMRG due to its quasi-1D nature. The calculation of the dynamical structure factor based on the dynamical DMRG is also standard. In the discussion part, the authors state again “investigating systems on finite lattices, especially with many-body interactions incorporated, is always a challenge...”. However, as the DMRG allows accessing very large system size (say up to thousands of sites) for (quasi-)1D systems, this is not an issue for the current calculation.

Overall speaking, the authors indeed provide faithful results, and this paper should be published somewhere. However, it is not of significant importance and novelty for publication on Nature Communications.

We thank the Referee for evaluating our work. In short, the Referee is raising two issues: (1) the novelty/importance of our findings and (2) how numerically challenging are the obtained results.

We strongly disagree with the conclusions of the Referee on both of these points. In our opinion, such an assessment of our work is arising from a misunderstanding of the Hamiltonian

studied in our manuscript and a misunderstanding of the computational technique used. We believe that the answers provided below, the new results included in our work, and the changes in the manuscript will dispel all doubts, and the Referee will consider our work as suitable for Nature Communications.

(1) The Referee suggests that one can reach a similar conclusion on the basis of Ref. 6 and Ref.7 [and another related work now included in the references, i.e., Phys. Rev. B **75**, 144420 (2007)]. We disagree with this statement. (i) In the aforementioned works, the stability of the Haldane phase was investigated within a two-leg ladder $S = 1/2$ Hubbard Hamiltonian with additional single diagonal hopping t_D , which (for large values) leads to the so-called bond-singlet phase (akin state to the ground state of the $S = 1$ Heisenberg model). In the other limit (large rung t_R), a gapped run-singlet phase is realized. The authors of that reference show that one can adiabatically connect these phases, “moving” between trivial and topologically nontrivial ground-states. See the sketch below for a pictorial description of the system:

Figure from Phys. Rev. B **75**, 144420 (2007)

Importantly, in the publication cited above, when changing the model parameters t_1, t_2, t_R, t_D and interaction U , the spin gap of the system is always finite. On the other hand, our results indicate a critical value of the interaction at which the spin gap opens. Already this result is novel and not discussed in the literature (to our best knowledge). More importantly, the adiabatic connection between the bond and rung singlet discussed in the works of F. Anfuso, A. Rosch, S. Moudgalya, and F. Pollmann is not relevant to our investigation. **It is impossible to adiabatically connect the gapless phase at $U, J_H \rightarrow 0$ with the gapped topologically nontrivial phase found at $U, J_H \gg t$ in our model.** Because the nature of this topological phase transition is the essence of our work, we believe that such results were not reported/investigated before.

(ii) The Referee writes that the models are “*only different at a Hund term, which should not affect on the presence of a topological transition*” when discussing our model and the models of Ref. 6 and Ref.7. This comment is a consequence of a severe misunderstanding of the model studied by us. In fact, without the Hund term, the system is always in a topologically trivial phase. In such limit ($J_H = 0$), our model can be described by two $S = 1/2$ Hubbard chains interacting only via the density-density term [the U' term in the second line of Eq.(1)].

Importantly, from the point of view of the two-orbital Hubbard model, such limit is **gapless**, as evident from the results presented in Fig.3D. **The finite Hund interaction is a necessary ingredient to stabilize the $S = 1$ Haldane phase in the two-orbital Hubbard model.** While visually, a two-orbital Hubbard model can be represented using a two-leg ladder, with one orbital per leg, the coupling between them primarily occurs via the Hund coupling, i.e., there are no hoppings between the effective legs.

(iii) The Referee is mistakenly assuming that equivalent terms analog to the rung hopping and/or diagonal hopping (as discussed in the aforementioned works) are also present in our system and that the Hund term is just an addition. This is incorrect. In the original version of the manuscript, we have considered degenerate bands; in the language of Ref. 6 and Ref. 7, we had considered $t_1 = t_2 = t = 0.5$ and $t_D = t_R = 0$. In such a limit, Ref. 6 and Ref. 7 predict a trivial phase, irrespective of the value of the interaction U .

One should also note that from the multi-orbital physics perspective, the finite t_R and t_D which are introduced in Refs. [6,7] are not physical. Different atomic orbitals have different symmetries (e.g., e_g orbitals), preventing direct (on-site; rung) kinetic terms. **There are no same-site hoppings in any realistic multi-orbital Hamiltonian.** A finite diagonal hopping t_D along only one direction is also not physical because the hybridization is typically symmetric, i.e., the hybridization term connects (site-1,orbital-1) with (site-2,orbital-2) but also should connect (site-1,orbital-2) with (site-2,orbital-1). Note that in such a setup, even a large value of hybridization does not lead to unpaired spins at the edge of the system, as required in the Haldane state.

(iv) Furthermore, in Ref. [7] [S. Moudgalya and F. Pollmann, Phys. Rev. B **91**, 155128 (2015)] it was argued that charge fluctuations will destroy the Haldane phase if reflection symmetry is broken (i.e., when $t_1 \neq t_2$). In our language, such a scenario corresponds to the case of non-degenerate bands. In order to strengthen our arguments, in the revised version of the manuscript, we consider two additional cases of kinetic energy. I.e., we consider non-degenerate bands and also hybridized orbitals, namely hopping matrices

$$t = t_{\gamma\gamma'} = \begin{pmatrix} -0.5 & 0 \\ 0 & -0.3 \end{pmatrix} \quad \text{and} \quad t = t_{\gamma\gamma'} = \begin{pmatrix} -0.5 & -0.5 \\ -0.5 & -0.3 \end{pmatrix},$$

respectively. In all considered cases, we found that the transition from the gapless phase at $U = J_H = 0$ to the topological Haldane phase occurs at $U \sim W$ (where W is the half-bandwidth of the kinetic energy of a given case, $W = 1, 1, 1.8$, respectively). See the new Fig. 6 and the new paragraph in the *Discussion and conclusion* section.

(2) Concerning the second point: how numerically challenging are our results? The Referee claims that *“the DMRG allows accessing very large system size (say up to thousands of sites) for (quasi-)1D systems, this is not an issue for the current calculation”*. We strongly disagree with this statement because it seems generic to any Hamiltonian, in particular ours.

Although it is possible to evaluate the quantum system of “*thousands of sites*”, this can be achieved with accuracy only for models of much smaller (local) Hilbert space, i.e., simple $S = 1/2$ Heisenberg Hamiltonian. Even then, the quantities of interest that can be accessed are the static observables or “just” the ground state energy and not the dynamical spectral functions considered in our work. Just to name a few examples of recent publications in which dynamical spectra were considered: $L = 80$ sites for the one-dimensional Hubbard model was presented in Nat. Commun. **14**, 3129 (2023); $L \sim 200$ sites for the $S = 1/2$ Heisenberg model was investigated in Phys. Rev. Lett. **127**, 03720 (2021) or Nat. Commun. **12**, 3599 (2021); $L = 64$ rungs of the two-leg Heisenberg ladder were considered in arXiv:2306.14742. Note that all of the above-mentioned works present state-of-the-art numerical investigations. All of these considerations work (effectively) with Hilbert space smaller than the one considered in our work. In the naive correspondence to the single-orbital Hubbard model, our two-orbital results are equivalent to calculations performed on 200 sites. For example, in Fig.2 (dynamical spin structure factor), we present results for $L = 80$ sites with a local Hilbert space of 16 states, Fig.4 (entanglement spectrum) with $L = 140$, or Fig.5 with $L = 100$ (static string order). In contrast, an akin model to our investigation, i.e., the SU(3) Fermi–Hubbard model, was recently investigated in Nat. Phys. **18**, 1201 (2022) on “just” $L = 30$ sites!

In summary, the Referee's statement is incorrect. The dynamical (energy-resolved) spectral functions are challenging even for a “simple” one-band Hubbard model. **The results for the two-orbital Hubbard model presented in our manuscript are state-of-the-art, and we are not aware of any other work that investigates such (large) systems.**

Reply to Referee #3:

This manuscript is devoted to a study of a one-dimensional model of interacting electrons that generalizes the Hubbard model to a two-orbital case. This is the simplest possible model with itinerant electrons that admits as a simple limit the spin-1 Heisenberg model. It thus allows to study the fate of the Haldane phase when electrons are itinerant. This is motivated by real magnets including Nickel ions for example as spin-1 localized moments. The authors use the DMRG algorithm to give an in-depth study of this model and they are able to characterize the extent and robustness of the Haldane phase. The results are interesting and obtained by state of the art theoretical analysis. I am inclined to recommend for publication in Nature Communications. However I am a bit disappointed by the following point: the authors claim to study in essence a more realistic depiction of real-world Haldane magnets - they correctly mention Nickel, Vanadium and Ruthenium ions - but their conclusions are seemingly only for hard-core theorists. No feedback is given concerning the impact of their results in present or future compound synthesis or experiments. I think that adding such arguments would strengthen the manuscript. I appreciate the fact that the manuscript is beautifully prepared, very well written and easy to read.

We thank the Referee for raising this issue since the revised version of the manuscript benefited strongly from their comments. To increase the impact of our results, we have performed additional calculations for various band structures (degenerate, non-degenerate, and hybridized orbitals). Consequently, we show that our results are relevant for various low-dimensional $S = 1$ compounds, irrespective of kinetic energy details, i.e., for recently investigated platforms such as metal-organic structures (Ref. 49) the van der Waals oxide dichlorides, e.g., OsCl_4 and RuOCl_2 (Ref. 48) as well as OsOCl_2 (Ref. 47). The three of these last materials were studied by members of this collaboration using *ab initio* techniques, and the hopping matrix is close to the unit matrix.

Specifically, in the previous version of the manuscript, we had considered only the degenerate bands, i.e., the hopping matrix of the form

$$t = t_{\gamma\gamma'} = \begin{pmatrix} -0.5 & 0 \\ 0 & -0.5 \end{pmatrix}.$$

In the new version of the manuscript, we vastly enlarged our scope by considering more realistic scenarios, i.e., the case of non-degenerate bands and hybridized orbitals,

$$t = t_{\gamma\gamma'} = \begin{pmatrix} -0.5 & 0 \\ 0 & -0.3 \end{pmatrix} \quad \text{and} \quad t = t_{\gamma\gamma'} = \begin{pmatrix} -0.5 & -0.5 \\ -0.5 & -0.3 \end{pmatrix},$$

respectively. In all these considered scenarios, we found that the transition from topological to the Haldane phase occurs at $U \sim W$ (where W is the half-bandwidth of the kinetic energy for

each given case, $W = 1, 1, 1.8$, respectively). See the new Fig. 6 and the new paragraph in the *Discussion and conclusion* section. Although the idealized case of degenerate bands is unlikely in real materials, these additional results prove that the phenomena discussed in our work are generic for one-dimensional two-orbital models and can be applied to various compounds, precisely as the referee wanted to see in the revised manuscript when writing “impact of their results in present or future compound synthesis or experiments”.

REVIEWERS' COMMENTS

Reviewer #1 (Remarks to the Author):

The authors have responded to my comments, and I am satisfied with the manuscript. Regarding the comments raised by the other referees:

It appears to me that the concerns of Referee 2 are successfully dispelled.

I agree with Referee 3 that the conclusions of the study is mainly aimed at theorists. The author's reply and modifications made in the revised text do not really address this issue. The paper could be strengthened by a discussion of experimental predictions or consequences of their findings.

Reviewer #2 (Remarks to the Author):

In this round of review, the authors present a comprehensive explanation regarding the novelty of their findings and the alignment of their model design with real-world scenarios. This information is helpful and addresses some of my questions. It would be highly appreciated if the authors could briefly include (some of) these discussions in the main text.

On the other hand, I still harbor doubts about the challenge of the numerical calculation and strongly recommend that the author revise the first paragraph of their discussion in the discussion and conclusion part. This will not diminish the significance of the current findings but will enhance the faithfulness of the discussions. So my opinion is to publish this paper on a specialized journal rather than Nat. Comm..

In addition, one minor suggestion is, the colors in Fig. 4(a) is not easy to distinguish for #3 and #4. They are somehow similar. I think the ES in the Haldane phase should be degenerated. It is better to represent #3 by a different color.

Reviewer #3 (Remarks to the Author):

The revised manuscript certainly answers many if not all my previous queries. The authors comments are also satisfactory. I recommend publication in its present version

Reviewer #4 (Remarks to the Author):

Reply to the Referee #1:

The authors have responded to my comments, and I am satisfied with the manuscript. Regarding the comments raised by the other referees:

It appears to me that the concerns of Referee 2 are successfully dispelled.

I agree with Referee 3 that the conclusions of the study is mainly aimed at theorists. The author's reply and modifications made in the revised text do not really address this issue. The paper could be strengthened by a discussion of experimental predictions or consequences of their findings.

Following the advice of the Referee, at the end of the “Discussion and Conclusion” section, we have included an additional paragraph on the experimental predictions or consequences of their findings. We have also listed another compound on which our results could be (in principle) tested.

Reply to the Referee #2:

In this round of review, the authors present a comprehensive explanation regarding the novelty of their findings and the alignment of their model design with real-world scenarios. This information is helpful and addresses some of my questions. It would be highly appreciated if the authors could briefly include (some of) these discussions in the main text.

Following the Referee's advice, we have rewritten the introduction part (first page, right column), emphasising the relevance of our model.

On the other hand, I still harbor doubts about the challenge of the numerical calculation and strongly recommend that the author revise the first paragraph of their discussion in the discussion and conclusion part. This will not diminish the significance of the current findings but will enhance the faithfulness of the discussions. So my opinion is to publish this paper on a specialized journal rather than Nat. Comm..

As the Referee requested, we have rewritten the beginning of the “Discussion and Conclusion” section.

In addition, one minor suggestion is, the colors in Fig. 4(a) is not easy to distinguish for #3 and #4. They are somehow similar. I think the ES in the Haldane phase should be degenerated. It is better to represent #3 by a different color.

We thank the Referee for this comment. In the revised version of the manuscript, the color in Figure 4 was changed.